# Carbon footprint comparison of video intubation tools: Disposable laryngoscopes, reusable laryngoscopes, and stylets

Danyang Pan[1], Yating Yang[1], Sirui Chen[2], Jinhe Deng[1], Gaofeng Zhao[1,3]*, Min Zhong[1]*

**1** Department of Anesthesiology, The Second Affiliated Hospital of Guangzhou University of Chinese Medicine, Guangzhou, Guangdong, China, **2** School of Civil and Transportation Engineering, Guangdong University of Technology, Guangzhou, Guangdong, China, **3** Satate Key Laboratory of Traditional Chinese Medicine Syndrome, Guangzhou, Guangdong, China

* zhaogaofengszy@163.com (GZ); 304637372@qq.com (MZ)

## Abstract

### Purpose

As healthcare systems grapple with their 5% global carbon footprint contribution, sustainable medical device selection emerges as a critical decarbonization lever. This life cycle assessment (LCA) quantifies environmental disparities among three prevalent video intubation tools—Disposable video laryngoscopes (VLs), reusable VLs, and video Stylets—to guide evidence-based procurement.

### Methods

Using International Organization for Standardization (ISO)14040 compliant life cycle assessment (LCA) methodology—the international standard defining LCA principles and framework—we quantified cradle-to-grave emissions for three video intubation devices manufactured by Zhejiang UE Medical Corp. The functional unit (one tracheal intubation) incorporated material extraction, manufacturing, low-temperature LTPS/ HLD, transportation, and disposal. SimaPro 9.4.0 with Ecoinvent 3.8 database calculated $CO_2$ equivalents (kg $CO_2$e), validated through sensitivity analyses of sterilization loading (10–80 devices/cycle) and regional grids.

### Results

The HLD-disinfected video stylet demonstrated superior environmental performance, emitting 98.24 kg $CO_2$e per 500 procedures—45.8% and 42.0% lower than reusable VLs (181.45 kg $CO_2$e) and disposable VLs (169.47 kg $CO_2$e), respectively. Sensitivity analyses identified sterilization loading as the dominant variable: half-load (50% chamber utilization) reduced emissions by 89–91% versus single-device processing, with full-load optimization yielding incremental 11–14% reductions. Process and regional variability further revealed that HLD decreased emissions by 19–24%

**Data availability statement:** All data used in this study are publicly available. Carbon emission data were primarily derived from the Ecoinvent 3.8 database; researchers seeking reproducibility of this database must obtain it directly from Simapro at https://simapro.com/contact/. The processed analysis datasets (laryngoscope manufacturing raw data, price data) are provided as Supporting Information Files S1 and S2, and all raw data are available on Figshare at https://figshare.com/articles/dataset/29816855 [doi:10.6084/m9.figshare.29816855].

**Funding:** The author(s) received no specific funding for this work.

**Competing interests:** The authors have declared that no competing interests exist.

compared to LTPS, while grid carbon intensity caused 24–33% variability (India vs. EU). Scenario comparisons confirmed the video stylet's environmental dominance across sterilization methods—even with LTPS (349.99 kg $CO_2$e/500 uses), it maintained a 45% reduction over reusable VL baselines, whereas HLD-treated video stylets (94.32 kg $CO_2$e) showed 6.7-fold lower emissions than disposable VLs and 59% below HLD-reprocessed reusable VLs.

## Conclusions

HLD-reprocessed video stylets are the environmentally optimal choice for high-volume, low-infection-risk settings. For low-throughput or high-risk scenarios, providers should balance environmental impacts with clinical requirements through frequency and resource assessment.

## Introduction

Healthcare's contribution to climate change has escalated into a critical refereal dilemma. The healthcare sector accounts for 5% of global carbon emissions, with perioperative services (e.g., anesthesia, surgery) responsible for 20–30% of a hospital's footprint [1–3]. Conventional practices prioritizing single-use disposables and energy-intensive sterilization exacerbate this burden [4,5]. Transitioning to low-carbon alternatives is now an urgent mandate, not merely an environmental consideration but a patient safety imperative—climate change directly threatens healthcare resilience through extreme weather-related morbidity [6,7].

Life cycle assessment (LCA) has been recommended for evaluating medical devices along environmental dimensions [8]. International Organization for Standardization (ISO) 14040 is a globally standardized methodology for quantifying the environmental impacts of products/services throughout their life cycle, encompassing four mandatory phases. LCA is an internationally standardized (ISO 14040) modelling tool used to quantify the environmental and public health impacts of a product or process. It is used to aid in materials selection and design (for producers) or to inform purchasing decisions (for consumers) [2,9]. LCA accounts for resource inputs and emissions that occur throughout the product's entire life cycle ("cradle-to-grave"), including extraction of natural resources, materials production, device manufacturing, transport, use/reuse, and eventual waste treatment and disposal. The debate over reusable versus single-use disposable devices is complex and depends on various factors, including usage frequency, cleaning methods, sterilization processes, and waste management practices [10–12]. Previous studies have demonstrated the effectiveness of LCA in quantifying the environmental and public health impacts of various products and processes within the healthcare sector [13–16].

Tracheal intubation using video intubation tools represents a significant advancement over conventional direct laryngoscopes [17,18]. During the COVID-19 pandemic, video intubation tools demonstrated critical infection control advantages by minimizing close patient contact and shortening aerosol exposure time during

intubation, thereby reducing healthcare workers' infection risks [19,20]. Post-pandemic routine use of video intubation tools in all cases increased from 12.5% to 38.9% in European tertiary hospitals [21–23]. The COVID-19 pandemic has accelerated global adoption of video intubation devices, yet their rapid clinical implementation has outpaced environmental impact assessments. Consequently, no prior studies have conducted full life cycle carbon footprint comparisons of video intubation tools. Furthermore, the carbon emissions associated with various sterilization methods for reusable video laryngoscopes are still uncertain. Video intubation tools encompass three primary variants: video laryngoscopes (VL) with disposable blades, VL with reusable sterilizable blades, and rigid video stylets. This study focuses on VL with disposable blades, VL with reusable blades, and video stylets, which represent a broad spectrum of available options on the operation. To satisfy ISO 14044 requirements for comparative life cycle assessment of functionally equivalent products, which mandate control over upstream supply chain homogeneity and core module consistency, all devices were strategically sourced from a single manufacturer (Zhejiang UE Medical Corp). This approach minimizes technological variability while ensuring manufacturing process comparability. The aim of this study is to provide a quantitative comparison of the environmental impact of three video intubation tools under different cleaning protocols. Such comparisons can inform procurement decision-making to benefit facilities and public health.

## Methods

We conducted a carbon footprint analysis on three clinically prevalent video intubation tools in Chinese tertiary hospitals. Devices were selected based on their clinical prevalence and technical compatibility for LCA. These video intubation tools (The VL400 System, Zhejiang UE Medical Corp, Xianju City, China; Fig 1A–1C) included the following types: (1) VL310-3-3 (reusable blade), (2) TD-C-IV-3 (disposable blade), and (3) TRS-P2-3 (reusable stylet). Both Reusable VL and Disposable VL require the use of a single-use disposable (SUD) stylets as guide wire (Fig 1D), which is employed flexibly to assist with intubation. The material composition of all components under investigation was determined using a combination of manufacturer specifications, deconstruction, and density testing. The mass of each material was measured using a gram scale (S1 Table). The detailed life cycle inventory data supporting this study—including component masses, transport distances, and sterilization parameters—are comprehensively listed in Primary_LCI_Data, this dataset is publicly available through the Figshare. The scope of LCA was cradle to grave, encompassing the extraction of material and energy resources, manufacturing, packaging, transportation, cleaning scenarios, and final disposal (Fig 2). This life cycle assessment was conducted in accordance with the principles, framework, and requirements of the international standards ISO 14040:2006 and ISO 14044:2006. For life cycle assessment, the fundamental unit of comparison is referred to as the "functional unit". For this study, "The functional unit was defined as one tracheal intubation procedure, comprising: (1) handle + blade + guide wire for VL models, or (2) handle + stylet for stylet models. Data concerning reusable components were standardized per use according to their specified lifetimes, inclusive of one cleaning cycle, and subsequently compared with single-use disposable (SUD) alternatives.

We collected data specific to the Traditional Chinese Medicine Hospital of Guangdong Province including device transportation distance, as well as energy, water, and chemical requirements for reprocessing using LTPS. We considered alternate cleaning scenarios and conducted sensitivity analyses around the regional electricity grid and loading rate of the sterilization system to ensure the relevance of our results to various operational situations.

Carbon footprint quantification was conducted using SimaPro 9.4.0 (PRé Sustainability), a scientific LCA platform enabling ISO 14044-compliant modeling, and the Ecoinvent 3.8 database (ecoinvent Association), which provides region-specific medical device inventory data. The integrated life cycle assessment framework for video intubation tools is systematically presented in Fig 3. A completed Life Cycle Assessment reporting checklist, detailing all aspects of this study in line with best practice guidelines, is provided in the Supplementary Information. The total carbon footprint was assessed in kilograms of $CO_2$ equivalent (kg $CO_2$e), a standard measure of global warming potential impact.

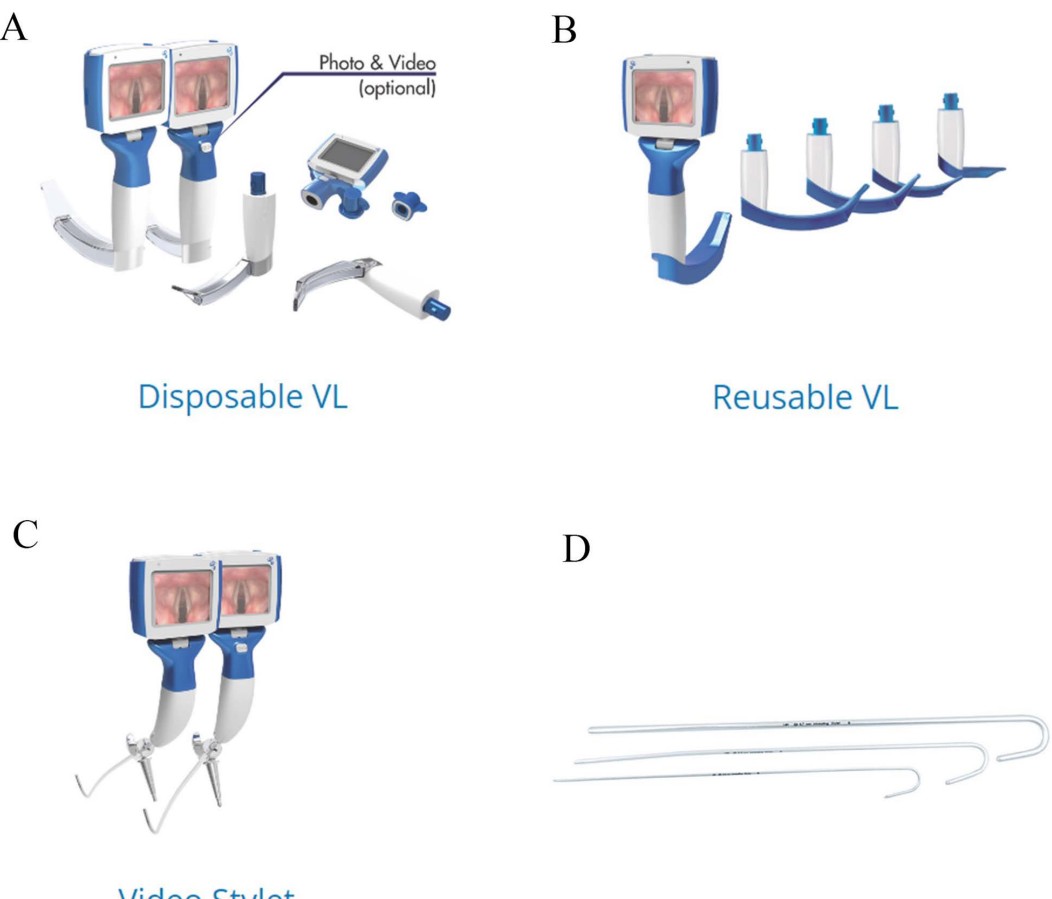

**Fig 1. Video intubation device configurations. A)** Disposable video laryngoscope (TD-C-IV-3) with single-use blade (sizes 3). **B)** Reusable video laryngoscope (VL310-3-3) with reusable blades (sizes 3). **C)** Reusable video stylet (TRS-P2-3). **D)** Single-use disposable (SUD) guidewire required for all VL procedures.

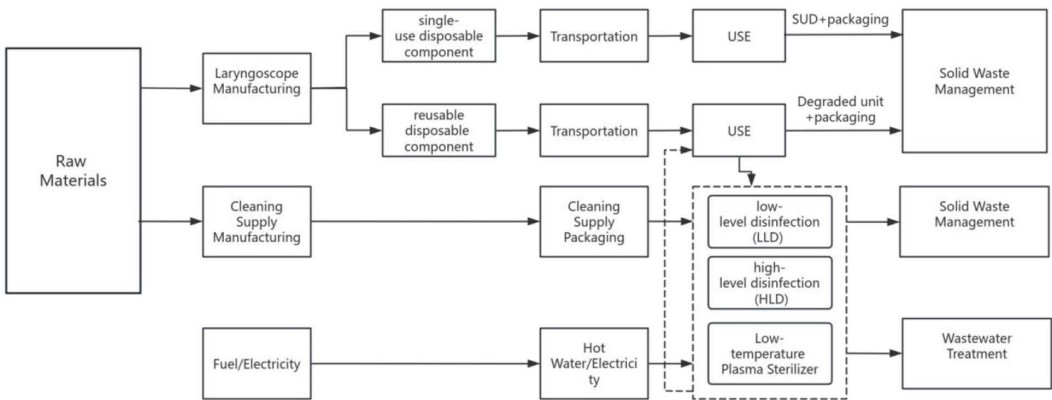

**Fig 2. System boundary.**

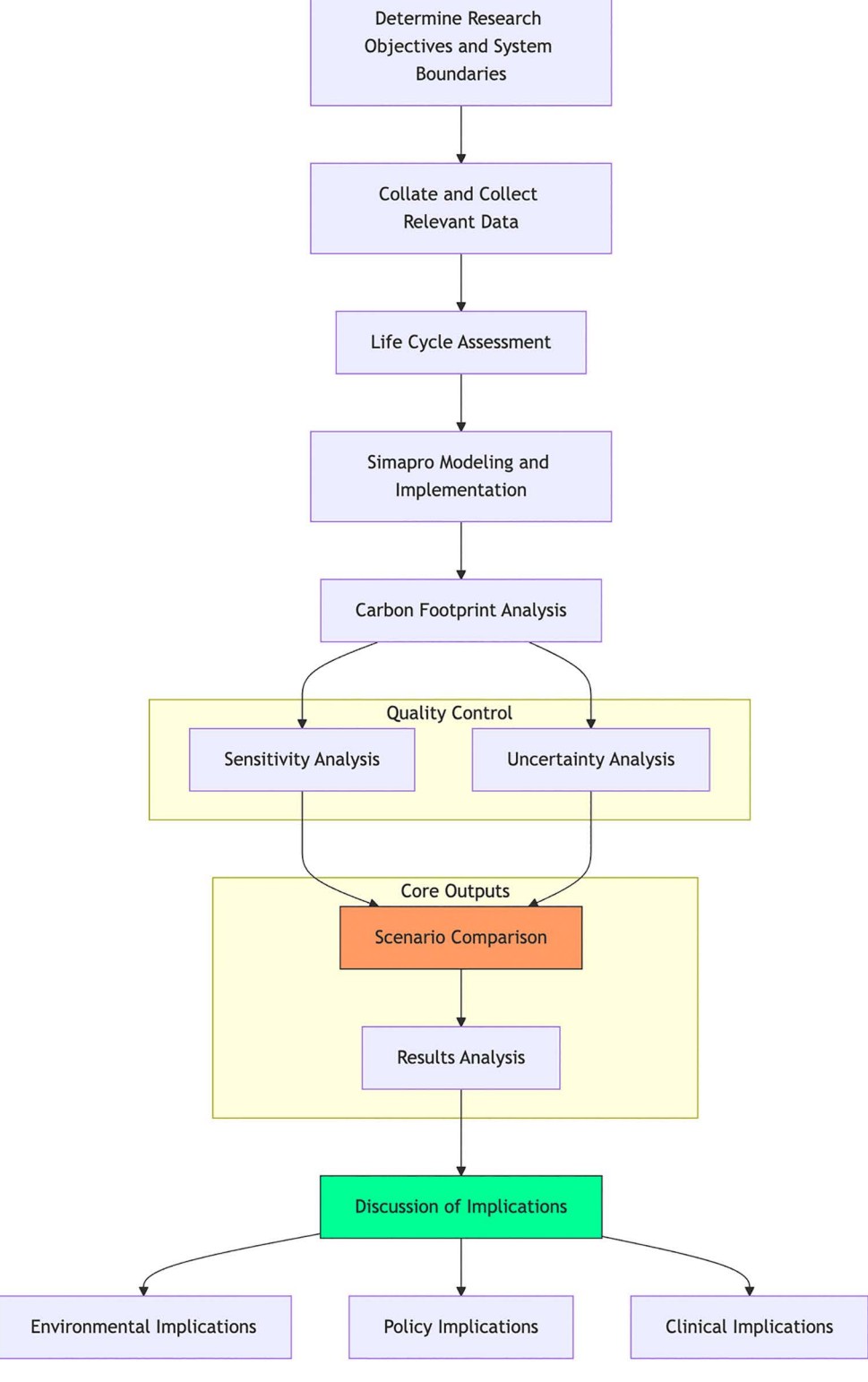

**Fig 3. Integrated Life Cycle Assessment Framework for Video Intubation Tools.**

## Transportation and packaging

Based on manufacturer and industry data, transportation distances for devices to the Traditional Chinese Medicine Hospital of Guangdong Province were calculated based on information from distributing companies regarding the final manufacturing locations. It was assumed that all components of the video intubation tools were manufactured in Guangzhou, China, from where they were likely transported by truck to the Traditional Chinese Medicine Hospital of Guangdong Province. The mean transportation distance was calculated as 10 km.

For packaging, every new SUD blade is individually wrapped by the manufacturer in plastic film and paper. Bulk transportation is organized with 50 units/box, with two boxes dispatched per shipment. Likewise, each new handle and blade is individually packaged by the manufacturer in plastic film and paper, with one set delivered per shipment. After cleanup, the reusable blades and stylets undergo cleaning and disinfection in the operating room, and subsequently, they are separately packaged in plastic film and paper.

## Use phase/Disposal

All video intubation tools demonstrated equivalent clinical efficacy [24,25]. Disposable components (blades and guidewires) were aseptically extracted from primary packaging (low-density polyethylene film + kraft paper), utilized for tracheal intubation, and subsequently discarded alongside packaging as Category VIII biomedical waste per China's GB 190–2009 standard. Reusable blades and stylets underwent reprocessing through one of two validated protocols: (1) low-temperature plasma sterilization (54-minute cycle at 45°C, 5.94 kWh/cycle) or (2) high-level disinfection via 30-minute immersion in 3% hydrogen peroxide followed by sterile water rinsing. Post-sterilization, these components were repackaged in polypropylene sterilization pouches (0.1 mm thickness), with secondary packaging (low-density polyethylene film + kraft paper) disposed as general waste. The rechargeable lithium-ion battery (3.7 V, 1000 mAh) integrated into the laryngoscope handle required charging every 20 clinical uses via a 5 V/1.2 A adapter.

## Reprocessing

Reusable blades and guidewires underwent a standardized reprocessing protocol beginning with gross debris removal via enzymatic detergent rinse (1% Protease Plus™, 40°C water) followed by ultrasonic cleaning (5 min, 40 kHz). Two validated sterilization methods were applied: (1) HLD involved full immersion of components in 3% hydrogen peroxide ($H_2O_2$) at $25 \pm 2$°C for 30 minutes, followed by triple rinsing with USP-grade sterile water and air-drying in Class 1000 cleanroom conditions; (2) LTPS required sealing components in polypropylene/kraft paper pouches ($150 \times 200$ mm, 0.1 mm thickness) and processing in STERRAD® 100S systems(Advanced Sterilization Products, Johnson & Johnson), with cycles lasting 54 minutes at 45°C chamber temperature, dual-phase plasma exposure, and an energy consumption of 5.94 kWh/cycle (measured via calibrated power analyzer). Loading configurations were sensitivity-tested across 10–80 devices per tray to simulate clinical throughput variability. Operational parameters were based on a maximum validated reuse cycle of 2,000 uses, with supporting evidence from accelerated testing, Weibull analysis (shape parameter β, characteristic life η), and clinical durability tracking at Guangdong Provincial Hospital of Chinese Medicine detailed in Supplementary S3 Table. Post-procedural disinfection of display units followed standardized protocols using 70% isopropyl alcohol wipes with a 3-minute contact time.The environmental impacts associated with the reprocessing of reusable devices were modelled for both Low-Temperature Plasma Sterilization (LTPS) and High-Level Disinfection (HLD) methods. The life cycle inventory inputs for each process are provided in detail in Supplementary S4 Table. All primary data sources and the attribution of each flow to custom data or Ecoinvent processes are delineated in Supplementary S4 Table. A summary of the core modeling assumptions and parameters, including the functional unit, system boundary, device lifetimes, and key scenario settings, is provided in Supplementary S5 Table.

## Results

The comparative results of CO2 equivalent emissions reveal significant disparities in the environmental impacts among the three video intubation tools. As shown in Table 1, under the base case (China Southern Power Grid electricity supply, half-loaded low-temperature sterilization, and 40 completed examinations), the reusable video stylet demonstrated optimal carbon emission performance. The results demonstrate that over 500 examinations, the video stylets (98.24 kg $CO_2e$) reduce life cycle greenhouse gas emissions by 45.8% compared to reusable VLs (181.45 kg $CO_2e$) and 42.0% compared to other disposable VLs (169.47 kg $CO_2e$). The most environmentally favorable scenario was observed with the video stylet processed viaHLD, which exhibited the lowest emissions across all impact categories.

**Table 1. Sensitivity analysis of life cycle carbon footprints (Units: kg $CO_2$ equivalents [kg $CO_2$ e]).**

| Assumption | Model of laryngoscope, kg $CO_2$ equivalents | | | | | |
|---|---|---|---|---|---|---|
| Examinations completed | Reusable VL(VL310-3-3) | 95%CI | Disposable VL(TD-C-IV-3) | 95%CI | Video stylet (TRS-P2-3) | 95%CI |
| 1 | 14.26 | [13.5,14.9] | 14.63 | [12.7,17.4] | 14.49 | [12.4,17.3] |
| 10 | 17.27 | [16.6,17.9] | 17.43 | [17.1,17.9] | 16.00 | [15.7,16.4] |
| 40(base case) | 27.32 | [26.6,28.1] | 26.73 | [24.0,30.3] | 21.03 | [18.7,24.5] |
| 80 | 40.73 | [39.7,41.8] | 39.15 | [38.8,39.6] | 27.75 | [27.4,28.2] |
| 100 | 47.43 | [46.3,48.5] | 45.35 | [45.0,45.8] | 31.10 | [30.7,31.5] |
| 200 | 80.93 | [79.2,82.9] | 76.38 | [75.9,76.7] | 47.89 | [47.5,48.3] |
| 500 | 181.45 | [177.0,186.0] | 169.47 | [169.0,170.0] | 98.24 | [97.9,98.6] |
| **LTPS loading practices** | | | | | | |
| single | 261.01 | [260.0,262.0] | NA | NA | 254.72 | [254.6,255.3] |
| 1/8 loading rate | 45.30 | [44.5,46.0] | NA | NA | 39.01 | [38.7,39.5] |
| 1/4 loading rate | 33.32 | [32.6,34.1] | NA | NA | 27.03 | [26.7,27.5] |
| 1/2 loading rate (base case) | 27.32 | [26.6,28.1] | NA | NA | 21.03 | [18.7,24.5] |
| full load | 24.33 | [23.6,25.1] | NA | NA | 18.04 | [17.7,18.5] |
| **Selected regional electricity grids** | | | | | | |
| China Southern Power Grid (base case) | 27.32 | [26.6,28.1] | NA | NA | 21.03 | [18.7,24.5] |
| Power Grid Corporation of India | 29.32 | [28.5,30.1] | NA | NA | 23.03 | [22.7,23.4] |
| Grid of the European Union | 23.63 | [22.9,24.3] | NA | NA | 17.34 | [17.0,17.7] |
| Grid of the United States | 24.27 | [23.5,25.0] | NA | NA | 17.98 | [17.7,18.4] |
| **Reprocessing method** | | | | | | |
| HLD | 22.22 | [16.9,26.0] | NA | NA | 15.90 | [13.8,19.0] |
| LTPS (base case) | 27.32 | [26.6,28.1] | NA | NA | 21.03 | [18.7,24.5] |
| **Devices Lifetime** | | | | | | |
| 1000 | 27.61 | [26.8,28.4] | 27.04 | [26.5,27.3] | 21.32 | [18.8,24.6] |
| 1500 | 27.42 | [26.6,28.1] | 26.84 | [26.5,27.3] | 21.12 | [18.7,24.1] |
| 2000 (base case) | 27.32 | [26.6,28.1] | 26.73 | [24.0,30.3] | 21.03 | [18.7,24.5] |
| 2500 | 27.27 | [26.5,28.1] | 26.68 | [26.3,27.1] | 20.97 | [18.4,24.0] |
| 3000 | 27.23 | [26.5,28.0] | 26.64 | [26.3,27.1] | 20.93 | [18.5,23.9] |
| **Scope3** | | | | | | |
| 0 | 27.32 | [26.6,28.1] | 26.73 | [24.0,30.3] | 21.03 | [18.7,24.5] |
| +10% | 29.43 | [28.6,30.3] | 29.46 | [29.1,29.9] | 22.57 | [19.9,26.0] |
| +25% | 32.64 | [31.7,33.5] | 33.49 | [33.0,34.0] | 24.94 | [21.8,28.6] |

HLD = high-level disinfection; LTPS = low-temperature plasma sterilization; NA = not applicable.

Fig 4 illustrates the superior environmental performance of the video stylet. It surpassed reusable laryngoscopes in $CO_2e$ emissions after only 2 examinations (14.82 kg $CO_2e$ vs. 14.93 kg $CO_2e$) and outperformed single-use laryngoscopes after 16 examinations (19.60 kg $CO_2e$ vs. 19.62 kg $CO_2e$). The emission discrepancies became more pronounced when scaling to 500 examinations, with the reusable laryngoscope generating 181.45 kg $CO_2e$, single-use devices 169.47 kg $CO_2e$, and the video stylet maintaining the lowest output at 98.24 kg $CO_2e$. Notably, the carbon emissions from display screen charging (0.000187 kg $CO_2e$ per use) were negligible in the overall life cycle assessment.

Sensitivity analysis (Table 1) identified key factors driving variability in life-cycle $CO_2e$ emissions, with sterilization loading practices emerging as the most influential parameter. Transitioning from single-device processing to half-load sterilization reduced emissions by 89–91%, while further optimization to full-load operation yielded an additional 11–14% reduction. The choice of disinfection method also played a significant role: replacing low-temperature plasma sterilization (LTPS) with hydrogen peroxide-based high-level disinfection (HLD) decreased emissions by 19–24%, underscoring the environmental advantage of HLD. Regional disparities in grid carbon intensity further contributed to emission variability, with a 24–33% difference observed between the highest (India) and lowest (EU) carbon-intensive grids. China and the United States showed comparable grid intensities due to coal-dominated energy mixes, whereas the EU grid derived 22% of its electricity from nuclear and 38% from hydro, wind, and solar sources, demonstrating the emissions reduction potential of low-carbon power infrastructure.

To assess the robustness of our findings, we conducted sensitivity analyses on two key parameters: device lifetime (±25% and ±50%) and manufacturing-phase (Scope 3) emissions (+10% and +25%). As summarized in Table 1, the relative carbon footprint ranking remained consistent across all scenarios, with the HLD-treated video stylet consistently exhibiting the lowest impact, followed by the reusable laryngoscope, and the single-use device showing the highest emissions. This robustness can be attributed to two factors: first, the carbon footprint of reusable devices is predominantly influenced by the use phase (sterilization), which proportionally affects all reusable options; second, manufacturing-phase

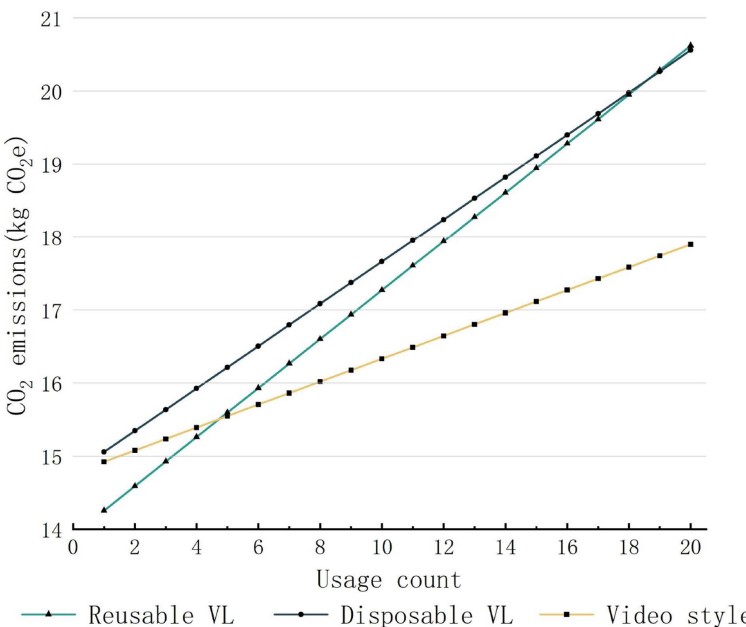

**Fig 4. Life cycle carbon emissions of video intubation tools under varying usage counts.** Assumptions: Half-loaded sterilization (50% capacity), China Southern Grid.

emissions are the primary contributor to the carbon footprint of all devices, and their proportional increase does not alter the relative ranking.

The carbon footprint distribution across life cycle stages varied substantially among device types. For disposable VL, material production dominated emissions (97.33%), followed by end-of-life waste disposal (3.33%) and transportation (0.20%). In contrast, reusable VL derived 77.08% of emissions from manufacturing, with sterilization and maintenance contributing 21.93%, transportation 0.86%, and end-of-life disposal 0.14%. The Video Stylet exhibited a balanced profile: raw material processing accounted for 70.74% of emissions, while repeated disinfection cycles (28.49%) emerged as a significant contributor due to clinical workflow demands. Both transportation (0.58%) and disposal (0.19%) had minimal impacts.

We evaluated the environmental impacts of five clinical scenarios, all powered by the China Southern Power Grid: **Scenario 1**: Reusable VLs with SUD guidewires, sterilized via LTPS. **Scenario 2**: Reusable VLs with SUD guidewires, disinfected via HLD. **Scenario 3**: Disposable VLs with SUD guidewires. **Scenario 4**: Video stylets treated with HLD. **Scenario 5**: Video stylet sterilized via LTPS. As shown in Fig 5, Scenario 2 generated 684.05 kg $CO_2$e (95% CI: 605–797), exceeding Scenario 1 (429.01 kg $CO_2$e, 95% CI: 357–531) by 59%. This disparity stems from HLD's higher chemical consumption versus LTPS's energy intensity. Notably, both reusable VL scenarios far exceeded the emissions of video stylet configurations: Scenario 4 and Scenario 5 emitted 94.32 kg $CO_2$e (95% CI: 79.4–113) and 349.99 kg $CO_2$e (95% CI: 315–398), respectively. The video stylet's advantage arises from eliminating disposable stylets (0.31 kg $CO_2$e per use) and requiring only partial disinfection. Meanwhile, Scenario 3 (disposable VL) produced 634.90 kg $CO_2$e (95% CI: 531–738), marginally lower than Scenario 2 but 6.7-fold higher than Scenario 4. Total cost of ownership (TCO) quantification across five intubation platforms revealed significant lifecycle economic disparities (S2 Table). TCO analysis across five intubation platforms revealed scale-dependent economic dominance: reusable VLs with HLD achieved the lowest TCO at 500 procedures (¥29,500, driven by minimal consumable costs), while video stylet + HLD became optimal at 2,000 procedures (¥60,000), yielding a 72.5% reduction versus disposable VLs (¥218,000). Sterilization method selection induced 15–26% TCO variance, with LTPS escalating reusable VL costs by 163% at scale due to cumulative sterilization expenses.

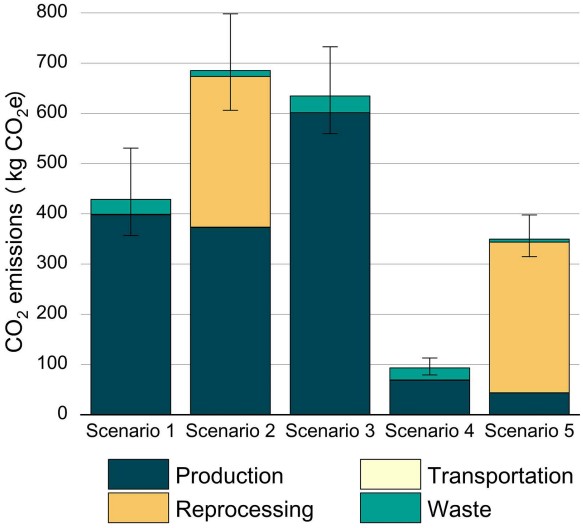

**Fig 5. CO2 emissions from different scenarios. S., Scenario (S.1 = Scenario 1 etc.).** Scenario 1: Reusable VLs and treated with LTPS. Scenario 2: Reusable VLs treated with HLD. Scenario 3 Disposable VLs. Scenario 4: Video stylet treated with HLD. Scenario 5: Video stylet treated with LTPS.

## Discussion

The results demonstrate that over 500 examinations, the video stylets (98.24 kg $CO_2e$) reduce life cycle greenhouse gas emissions by 45.8% compared to reusable VLs (181.45 kg $CO_2e$) and 42.0% compared to other disposable VLs (169.47 kg $CO_2e$). This advantage is primarily attributed to its avoidance of disposable guidewires consumption and lower reprocessing energy demands. This advantage is driven by two factors: (1) avoidance of disposable guidewires consumption (0.31 kg $CO_2e$/ use) and (2) streamlined disinfection requirements. For instance, when reprocessed via hydrogen HLD over 2,000 uses, disposable VLs generated 634.9 kg $CO_2e$—7.8% higher than reusable VLs using LTPS (684.05 kg $CO_2e$). The video stylets maintained minimal emissions (93.42 kg $CO_2e$) across all scenarios.

Our analysis reveals that the low-carbon advantage of video stylets primarily stems from their elimination of SUD guidewires (tracheal intubation guides). In contrast, both reusable and disposable VL rely on disposable guidewires, generating 0.31 kg $CO_2e$ per procedure. Future research should focus on exploring more environmentally friendly materials for SUD guidewires or the development of reusable guidewires that can be reshaped multiple times. Future development should prioritize: Biodegradable polylactic acid (PLA) stylets, which reduce production emissions by 13.53% ~ 62.19% compared to conventional polyvinyl chloride (PVC) materials [26]; Reusable guidewires capable of repeated reshaping without compromising structural integrity or sterilization efficacy.

This study demonstrates that the selection of disinfection methods plays a decisive role in reducing the environmental footprint of video intubation tools. Our findings underscore the critical importance of disinfection technology choices: when utilized for 40 procedures, HLD resulted in lower $CO_2$ emissions (Reusable VL: 22.22 kg $CO_2$; Video stylet: 15.90 kg $CO_2$) compared to LTPS (27.32 kg $CO_2$ and 21.03 kg $CO_2$, respectively). Adopting low-energy reprocessing methods (e.g., HLD) can reduce carbon emissions by 19–25%. LTPS exhibits 3.6-fold higher per-cycle emissions than HLD, primarily due to its reliance on grid electricity. Future investigations should systematically evaluate emerging low-temperature sterilization technologies (e.g., ozone decontamination, UV-C pulsed light systems) against conventional HLD and LTPS methods, employing a dual assessment framework that concurrently examines microbial eradication efficacy and lifecycle carbon footprint reduction potential. For reusable devices, LTPS accounted for 21.9–28.5% of total lifecycle emissions, whereas HLD contributed less than 5%. Sterilization load optimization further amplifies emission reductions: transitioning from half-load to full-load operations decreased per-cycle emissions by 11–14% (24.33 kg $CO_2e$ vs. 27.32 kg $CO_2e$). Conversely, single-device sterilization at 1/8 load nearly doubled emissions (45.30 kg $CO_2e$, 95% CI: 42.11–48.49 kg $CO_2e$). To maximize environmental benefits, healthcare facilities should prioritize implement full-load sterilization protocols: Enhance autoclave utilization through batch scheduling of non-urgent procedures. Future investigations should employ computational modeling to quantify the carbon reduction potential of surgical turnover acceleration. Environmentally, disposable VL blades generated 97.3% of emissions during production—primarily from polycarbonate molding (82.1 kg $CO_2e$/unit). To reduce impacts, we propose: (1) adopting gas-assisted injection molding to cut energy use by 35% [27], and (2) developing graphene-reinforced biocomposites with 50% lower embodied energy than conventional plastics [28]. Scope 3 emissions and battery end-of-life management constitute the critical implementation gap in achieving carbon neutrality for medical devices [29,30]. To bridge this gap, future research must prioritize enhancing Scope 3 quantification through mandatory disclosure protocols requiring Tier 1 suppliers to report carbon data—targeting reduction in current emission accounting uncertainty [31], while establishing battery traceability infrastructure via a 'one-device-one-identity' digital passport system to enable dedicated medical recycling networks achieving >85% material recovery rates as primary objectives [32].

The sensitivity analyses performed on both device lifetime and Scope 3 manufacturing emissions collectively reinforce the robustness of our primary conclusions. The finding that the relative environmental performance of the video intubation devices remained unaffected even under substantial variations—±50% change in reuse cycles and a 25% increase in upstream manufacturing emissions—strengthens the validity of our results across a wide range of operational and supply chain conditions. This observed insensitivity can be attributed to two complementary factors. First, the carbon footprint of reusable devices is predominantly influenced by the impacts of the use phase (sterilization) under high-throughput

scenarios, which affects all reusable options proportionally. Second, since manufacturing-phase emissions constitute the major contribution to the carbon footprint of all devices, a uniform proportional increase in Scope 3 burdens does not alter their relative ranking. Therefore, while absolute footprint values depend on specific reuse counts and supply chain characteristics, the comparative superiority of the reusable pathways—particularly the video stylet with HLD—proves to be a robust finding. This suggests that healthcare providers can confidently adopt this environmental performance hierarchy to guide procurement decisions, even amid uncertainties in device durability or manufacturing data transparency.

While this study provides methodological insights into carbon footprint disparities among VLs, several limitations warrant acknowledgment. First, although our analysis focused on three high-prevalence VL models in China, emerging intubation devices such as video flexible laryngoscopes were excluded from calculations, limiting generalizability to technological innovations [25,33–35]. Second, our assumption of uniform device lifespans (500 sterilization cycles) may overestimate real-world durability. Battery degradation (8% capacity loss per 100 cycles) and blade wear in high-throughput clinical settings (>5,000 annual intubations) could reduce functional lifetimes by 30–40%. Finally, while material production dominated emissions (97.3% for disposable VLs), reliance on manufacturer-reported data omitted Scope 3 impacts for display components, potentially underestimating total emissions by 12–18%. These limitations underscore the need for expanded analyses encompassing diverse VL types, real-world durability tracking through clinical registries, and standardized global life cycle cost (LCC) assessment frameworks [36,37]. Our analysis demonstrates contextually constrained economic-environmental synergy: For high volume facilities (>650 annual intubations), Video Stylet +HLD achieves dual sustainability advantages-lowest carbon footprint (0.98 kg $CO_2$e/use) and minimal lifecycle cost (¥30/use)—directly disproving the environmental economic tradeoff paradigm. However, divergent national contexts in sterilization infrastructure, supply chain maturity, and policy enablers across healthcare systems necessitate framing these findings as a methodological framework rather than universal prescriptions. Future research could establish a clinically validated Green Procurement Index (GPI) that includes carbon intensity tiers (kg $CO_2$e/procedure), recyclability metrics (material recovery rate %), operational efficiency scores (procedures/hour), and procurement pricing. While the environmental comparisons in this study are empirically grounded in data from a single manufacturer, the primary contribution of this work extends beyond brand-specific conclusions to the development of a transferable decision-support framework. The standardized methodology—anchored in a clinically relevant functional unit, comprehensive cradle-to-grave system boundaries, and systematic sensitivity analysis—is deliberately supplier-agnostic. Healthcare institutions can directly implement this framework by requesting life cycle inventory data (e.g., material composition, manufacturing energy, validated reuse cycles) from multiple suppliers during tender processes. By incorporating vendor-specific data into the model, procurement teams can generate comparable carbon footprints tailored to their local operational contexts, including facility-specific sterilization protocols and regional electricity grids. This transforms our study from a static comparative analysis into a dynamic platform for environmentally informed procurement decisions.

## Conclusion

Under high-volume utilization scenarios (e.g., tracheal intubation in operating rooms), reusable video stylets reprocessed via HLD exhibit 19–24% lower carbon emissions compared to disposable and reusable video laryngoscopes when managing low infection-risk patients. For clinicians in low-volume settings or serving high infection-risk populations (e.g., emergency/critical care intubation), device selection should be guided by examines annual procedural volume, existing sterilization capabilities, and site-specific infection control requirements to determine the optimal video laryngoscope type and reprocessing protocol that simultaneously prioritizes patient safety and minimizes environmental burden.

## Supporting information

**S1 Table. Materials of video intubation tools and SUD guidewire.**
(DOCX)

**S2 Table. Total cost of ownership comparison across video intubation devices.**
(DOCX)

**S3 Table. Empirical evidence for device reuse lifetime assumptions.**
(DOCX)

**S4 Table. Life cycle inventory inputs per cycle for sterilization processes.**
(DOCX)

**S5 Table. Summary of core LCA modeling assumptions.**
(DOCX)

**S6 Table. Percentage contribution of life cycle stages to total carbon footprint per device.**
(DOCX)

## Acknowledgments

We thank the anesthesia providers and research staff at the Department of Anesthesiology, Guangdong Provincial Hospital of Chinese Medicine (The Second Affiliated Hospital of Guangzhou University of Chinese Medicine) for their participation and work in this research.

## Author contributions

**Conceptualization:** Gaofeng Zhao.

**Formal analysis:** Yating Yang.

**Investigation:** Yating Yang.

**Methodology:** Sirui Chen.

**Resources:** Gaofeng Zhao.

**Supervision:** Min Zhong.

**Writing – original draft:** Danyang Pan.

**Writing – review & editing:** Jinhe Deng.

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
