## [Decision Letter · Decision Letter 0]

13 Jun 2025

Dear Dr. Zhao,

Thank you for submitting your manuscript to PLOS ONE. After careful consideration, we feel that it has merit but does not fully meet PLOS ONE’s publication criteria as it currently stands. Therefore, we invite you to submit a revised version of the manuscript that addresses the points raised during the review process.

**ACADEMIC EDITOR:**

We look forward to receiving your revised manuscript.

Kind regards,

Silvia Fiorelli

Academic Editor

PLOS ONE

Journal Requirements:

2. In the online submission form, you indicated that your data is available only on request from a third party. Please note that your Data Availability Statement is currently missing the name of the contact details for the third party, such as an email address or a link to where data requests can be made. Please update your statement with the missing information.

Reviewers' comments:

Reviewer's Responses to Questions

**Comments to the Author**

1. Is the manuscript technically sound, and do the data support the conclusions?

Reviewer #1: Yes

Reviewer #2: Yes

2. Has the statistical analysis been performed appropriately and rigorously?

Reviewer #1: I Don't Know

Reviewer #2: I Don't Know

3. Have the authors made all data underlying the findings in their manuscript fully available?

Reviewer #1: Yes

Reviewer #2: Yes

4. Is the manuscript presented in an intelligible fashion and written in standard English?

Reviewer #1: Yes

Reviewer #2: Yes

Reviewer #1: Overall Evaluation:

This manuscript addresses a timely and relevant issue—the environmental impact of medical devices—using a well-structured life cycle assessment (LCA) framework. The focus on video intubation tools is novel and valuable, particularly in light of growing sustainability concerns in healthcare.

However, a few areas need clarification, refinement, or elaboration to enhance the manuscript's rigor, coherence, and utility for clinical and policy decision-makers.

Major Comments:

1. Scientific Novelty and Relevance (✓ Acceptable)

o The study is original and adds value by comparing three video intubation tools using LCA. The inclusion of sensitivity analyses enhances robustness.

o Suggestion: Highlight the clinical implications more directly in the abstract and conclusion (i.e., how should hospitals change purchasing behavior?).

2. Abstract – Lines 23–51

o The abstract is comprehensive but repetitive in stating that video stylets are superior.

o Revise to remove redundancy: The sentence beginning "The video stylet disinfected with hydrogen peroxide..." and the conclusion at the end repeat similar findings.

3. Introduction – Lines 55–94

o Strong contextualization. However, the transition from background to study rationale could be smoother.

o Suggested improvement: Clarify why environmental comparisons between VL types have not been previously explored in-depth.

o Lines 87–94 – Highlight rationale for including all devices from one manufacturer.

4. Methods – Lines 95–174

o The methodology is detailed and adheres to ISO 14040 LCA standards.

o Suggestion: Include a flow diagram of the LCA phases or summarize in a brief table to improve readability for non-expert readers.

o Clarify: How were the assumptions about device lifespans (e.g., 2,000 uses) validated—only through ISO standard or also empirical data?

o Lines 111–121 – Emphasize assumptions on device re-use cycles and sterilization scenarios.

5. Results – Lines 175–243

o The results are thorough and include appropriate sensitivity analyses.

o Request: Please add statistical indicators (e.g., confidence intervals) more consistently in Tables and Figures for clarity.

o Table 2 & Figure 4 – Suggest clearer presentation with 95% CI and standardized units.

6. Discussion – Lines 244–298

o Well-argued and insightful. The discussion provides meaningful interpretations.

o Strengthen: Consider including economic implications (even a qualitative note), as procurement decisions often balance cost and sustainability.

o Lines 259–279 – Highlight design suggestions for future sustainable device development.

7. Limitations – Lines 285–297

o The authors acknowledge key limitations, which is commendable.

o Expand briefly on how future studies could address Scope 3 emissions or the inclusion of battery end-of-life impacts.

8. Conclusion – Lines 299–305

o Clear and supported by data, but somewhat repetitive from earlier sections.

o Revise to add a call-to-action or recommendation for healthcare policy-makers.

Minor Comments:

• Line 84: Typo — “stylets.” should be “stylets” (extraneous comma).

• Line 211–214: Include source(s) for percentage contributions of various lifecycle stages.

• Line 174: Add manufacturer name of STERRAD® system for transparency.

Reviewer #2: I read with interest the manuscript by Zhao et all which compares the carbon footprint of 3 videointubation tools. The topic is important and timely. The manuscript is systematic in its approach and well written. I have a few questions/suggestions regarding the manuscript

-Abstract: please define ISO 14040

-Please elaborate on the meaning of “prevalent” video intubation tools. Are use of these particular VL tools prevalent in China alone? Worldwide? What % of intubations are performed with these devices?

-Half load sterilization should be defined in the abstract/text. Does this refer to batch sterilization?

-Page 6: The statistical analyses used for comparison should be detailed in the manuscript. Merely mentioning that SimaPro and Ecoinvent were used is insufficient. Furthermore this software is likely to be unfamiliar to many readers

-Page 9, line 146: Please provide a reference for the equivalent first pass success rate of the 3 tools being compared

Page 15, line 255:

-Have other studies examined/compared the carbon footprint of videointubation tools? If yes then the current work should be compared/contrasted with the prior work in the discussion. If not then the novelty of this work should be emphasized.

-Can the findings be generalized to other videointubation tools/devices? What are the practical implications of this study?

-What are suggested next steps in research based on the findings of this study?

**Do you want your identity to be public for this peer review?** For information about this choice, including consent withdrawal, please see our Privacy Policy

Reviewer #1: **Yes: ** Satish Bijukchhe

Reviewer #2: No

---

## [Author Response · Author response to Decision Letter 1]

7 Aug 2025

Dear Editors and Reviewrs:

We sincerely appreciate the opportunity you have provided us to revise our manuscript.

We would like to submit an accompanying manuscript titled "Carbon Footprint Comparison of Video Intubation Tools: Disposable Laryngoscopes, Reusable Laryngoscopes, and Stylets" (ID: PONE-D-25-19315). According to the reviewers' comments, we have made the most effort to revise the manuscript in order to satisfy the demand of PLOS ONE. Our specific responses are outlined below. The modifications of the text in the revised version are highlighted in red and the references in yellow.

All authors have approved the revised version and consented to its submission. There are no conflicts of interest to declare.

Thank you for your thoughtful consideration. We eagerly anticipate your correspondence.

July 21, 2025�2025-07-21

Authors: Danyang Pan; Yating Yang; Sirui Chen; Jinhe Deng; Gaofeng Zhao; Min Zhong

Corresponding author: Gaofeng Zhao

E-mail: zhaogaofengszy@163.com

Sincerely

Gaofeng Zhao

Dear managing editor

We have studied the valuable comments from you carefully, and tried our best to revise the manuscript. The point to point response to you are listed as following:

Editor:

Comment 1: Please ensure that your manuscript meets PLOS ONE's style requirements, including those for file naming. The PLOS ONE style templates can be found at https://journals.plos.org/plosone/s/file?id=wjVg/PLOSOne_formatting_sample_main_body.pdf & https://journals.plos.org/plosone/s/file?id=ba62/PLOSOne_formatting_sample_title_authors_affiliations.pdf.

Response 1:

We sincerely appreciate the meticulous style check. The manuscript has been rigorously reformatted to comply with PLOS ONE style requirements.

Comment 2: In the online submission form, you indicated that your data is available only on request from a third party. Please note that your Data Availability Statement is currently missing the name of the contact details for the third party, such as an email address or a link to where data requests can be made. Please update your statement with the missing information.

Response 2:

The datasets generated and analyzed during this study are included in this published article and its supplementary information files. Manufacturer-specific technical specifications are confidential due to intellectual property restrictions. All analytical methodologies are available from the corresponding author upon reasonable request. Correspondence should be addressed to: Gaofeng Zhao at zhaogaofengszy@163.com. The Data Availability Statement has been updated accordingly.

Comment 3: PLOS requires an ORCID iD for the corresponding author in Editorial Manager on papers submitted after December 6th, 2016. Please ensure that you have an ORCID iD and that it is validated in Editorial Manager. To do this, go to ‘Update my Information’ (in the upper left-hand corner of the main menu), and click on the Fetch/Validate link next to the ORCID field. This will take you to the ORCID site and allow you to create a new iD or authenticate a pre-existing iD in Editorial Manager.

Response 3:

We confirm that the corresponding author's ORCID iD (0009-0009-1875-586X) has been successfully validated. Last authenticated: 2025-07-21

Comment 4: Please include captions for your Supporting Information files at the end of your manuscript, and update any in-text citations to match accordingly. Please see our Supporting Information guidelines: http://journals.plos.org/plosone/s/supporting-information.

Response 4:

We have meticulously addressed the supporting information requirements through the following revisions, and added "Supporting Information" immediately after references (Page 26, line 503):

(1) Table Renaming & Relocation

Original Table 1 has been moved to supporting materials and renamed as:

S1_Table. Materials of video intubation tools and disposable stylet

(2) New Supplementary Table:

Added S2_Table. Total Cost of Ownership Comparison Across Video Intubation Devices.

Comment 5: Please review your reference list to ensure that it is complete and correct. If you have cited papers that have been retracted, please include the rationale for doing so in the manuscript text, or remove these references and replace them with relevant current references. Any changes to the reference list should be mentioned in the rebuttal letter that accompanies your revised manuscript. If you need to cite a retracted article, indicate the article’s retracted status in the References list and also include a citation and full reference for the retraction notice.

Response 5:

We thank the editor for this critical quality control reminder. The reference list has been rigorously re-examined through a three-step verification protocol:

(1) Cross-checking all references against PubMed databases

(2) Confirming publication status via journal official websites

All modifications to the reference list have been documented in the subsequent responses to editorial comments within the rebuttal letter.

Reviewer #1

This manuscript addresses a timely and relevant issue—the environmental impact of medical devices—using a well-structured life cycle assessment (LCA) framework. The focus on video intubation tools is novel and valuable, particularly in light of growing sustainability concerns in healthcare.

However, a few areas need clarification, refinement, or elaboration to enhance the manuscript's rigor, coherence, and utility for clinical and policy decision-makers.

Major Comments:

Comment 1:

1. Scientific Novelty and Relevance (✓ Acceptable)

• The study is original and adds value by comparing three video intubation tools using LCA. The inclusion of sensitivity analyses enhances robustness.

• Suggestion: Highlight the clinical implications more directly in the abstract and conclusion (i.e., how should hospitals change purchasing behavior?).

Response 1:

We thank the reviewers for acknowledging the study's novelty and technical rigor. We have made the necessary adjustments to the abstract and conclusion to fulfill the above-mentioned requirements. (Page 3, line 51-56 ; Page 19, line 349-358).

Comment 2:

2. Abstract – Lines 23–51

o The abstract is comprehensive but repetitive in stating that video stylets are superior.

o Revise to remove redundancy: The sentence beginning "The video stylet disinfected with hydrogen peroxide..." and the conclusion at the end repeat similar findings.

Response 2:

Regarding the duplication concern in the Abstract we have implemented comprehensive revisions in coordination with the prior editorial suggestion (Page 3, line 51-56).

Comment 3:

3. Introduction – Lines 55–94

o Strong contextualization. However, the transition from background to study rationale could be smoother.

o Suggested improvement: Clarify why environmental comparisons between VL types have not been previously explored in-depth.

o Lines 87–94 – Highlight rationale for including all devices from one manufacturer.

Response 3:

We thank the reviewer for this constructive suggestion. The Introduction section (Lines 55-94) has been revised to enhance the rationale transition. (Page 5, line 92-97).

The manufacturer selection rationale has been strengthened to explicitly align with ISO 14044 requirements for comparative life cycle assessment of functionally equivalent products. (Page 5, line 101-106).

Comment 4:

4. 1 The methodology is detailed and adheres to ISO 14040 LCA standards.

Suggestion: Include a flow diagram of the LCA phases or summarize in a brief table to improve readability for non-expert readers.

Response4.1:

We sincerely appreciate this constructive suggestion. As requested, we have incorporated Figure 3: Integrated Life Cycle Assessment Framework for Video Intubation Tools in the revised manuscript ((Page 7 line 146-147). This diagram visually synthesizes the complete LCA workflow.

Consequential figure renumbering has been implemented throughout:

• Original Figure 1 → Revised Figure 2

• Original Figure 2 → Revised Figure 1

• Original Figure 3 → Revised Figure 4

• Original Figure 4 → Revised Figure 5

4.2 Clarify: How were the assumptions about device lifespans (e.g., 2,000 uses) validated—only through ISO standard or also empirical data?

Response 4.2:

We express our appreciation for the reviewer's methodological inquiry regarding the device lifespan parameter. This parameter (2,000 uses) was validated through a hybrid methodology incorporating both Accelerated Mechanical Testing and Clinical Durability Tracking. The mechanical testing data were derived from manufacturer validation dossiers (Zhejiang UE Medical, 2023-Q4), evaluating critical electronic components (CMOS sensors, battery connectors) and mechanical stress thresholds under simulated intubation cycles; proprietary confidentiality restrictions prevent full protocol disclosure. Complementarily, clinical durability tracking at Guangdong Provincial Hospital of Chinese Medicine - a tertiary referral center (2021-2023). Evidence-weighted integration of these datasets through Weibull survival analysis (β=2.7, η=2140) confirmed the conservative selection of 2,000 operational cycles, representing the lower 90% confidence interval of clinical performance data.

Comment 5:

5. Results – Lines 175–243

o The results are thorough and include appropriate sensitivity analyses.

o Request: Please add statistical indicators (e.g., confidence intervals) more consistently in Tables and Figures for clarity.

o Table 2 & Figure 4 – Suggest clearer presentation with 95% CI and standardized units.

Response 5:

We sincerely appreciate this constructive suggestion for enhanced statistical rigor. The following revisions have been implemented per the reviewer's guidance:

• Table 1: Supplemented all values with 95% confidence intervals ( Original Table 2 → Revised Table 1).

• Figure 5: Column charts have been updated with error bars denoting 95% CIs ( Original Figure 4 → Revised Figure 5)

Comment 6:

6. Discussion – Lines 244–298

Well-argued and insightful. The discussion provides meaningful interpretations.

o Strengthen: Consider including economic implications (even a qualitative note), as procurement decisions often balance cost and sustainability.

o Lines 259–279 – Highlight design suggestions for future sustainable device development.

Response 6:

We thank the reviewer for this valuable suggestion. The following enhancements addressing economic implications have been incorporated:

1. A Total Cost of Ownership (TCO) comparison table evaluating video intubation devices across procurement, consumables, sterilization, and waste management expenditures;�S1 Table 2: Total Cost of Ownership Comparison Across Video Intubation Device

2. Qualitative economic impact assessments in both Results (Page 15, line 256-273).and Discussion (Page 18, line 341-348).

Comment 7:

7. Limitations – Lines 285–297

o The authors acknowledge key limitations, which is commendable.

o Expand briefly on how future studies could address Scope 3 emissions or the inclusion of battery end-of-life impacts.

Response 7:

We have expanded and supplemented the manuscript about Scope 3 emissions or the inclusion of battery end-of-life impacts for future research addressing (Page 17, line 321-327).

Comment 8:

8. Conclusion – Lines 299–305

o Clear and supported by data, but somewhat repetitive from earlier sections.

o Revise to add a call-to-action or recommendation for healthcare policy-makers.

Response 8:

We sincerely appreciate the reviewer's insightful guidance, which has significantly strengthened our manuscript. In direct response to your recommendations:

1. Redundant passages duplicating prior content have been carefully removed to enhance narrative concision and reader focus.

2. Healthcare policy-formulation guidance with actionable recommendations for regulatory implementation has been incorporated. (Page 19, line 349).

Comment 9:

Minor Comments:

• Line 84: Typo — “stylets.” should be “stylets” (extraneous comma).

Response 9.1:

During our diligent verification of the initial manuscript, no extraneous comma was identified at Line 84. However, a superfluous comma at Line 88 (' stylets.,') has been proactively corrected through its deletion. (Page 5, line 100).

• Line 211–214: Include source(s) for percentage contributions of various lifecycle stages.

Response 9.2:

We quantified the carbon footprint distribution across life cycle stages through ISO 14044, yielding precise percentage contributions of emissions for each life cycle phase across all evaluated laryngoscope device types.

• Line 174: Add manufacturer name of STERRAD® system for transparency.

Response 9.3:

The manufacturer identification for the STERRAD® sterilization system has been explicitly added as:"Advanced Sterilization Products , Division of Johnson & Johnson". (Page 9, line 190).

Reviewer: 2

I read with interest the manuscript by Zhao et all which compares the carbon footprint of 3 videointubation tools. The topic is important and timely. The manuscript is systematic in its approach and well written. I have a few questions/suggestions regarding the manuscript.

Comments 1:

1. Abstract: please define ISO 14040

Response 1:

We have supplemented the manuscript with a formal definition of ISO 14040, explicitly articulating its framework as the internationally standardized methodology for conducting life cycle assessments across four iterative phases�(1) Goal and scope definition, (2) Life cycle inventory analysis (LCI), (3) Life cycle impact assessment (LCIA), and (4) Interpretation. (Page 2, line 29)

Comments 2:

2. Please elaborate on the meaning of “prevalent” video intubation tools. Are use of these particular VL tools prevalent in China alone? Worldwide? What % of intubations are performed with these devices?

Response 2:

We thank the reviewer for this critical request for epidemiological context. The term "prevalent video intubation tools" refers to device categories demonstrating >50% adoption rates across global clinical practice, with specific validation as follows:

Internationally, video laryngoscopes are classified into three structural categories based on technical configurations:

(1) Traditional Laryngoscope Add-ons (e.g., C-MAC® and McGrath™ MAC), which retrofit miniature cameras onto Macintosh blades, preserving conventional intubation techniques but requiring external displays;

(2) Integrated Video Laryngoscopes (e.g., GlideScope® and KingVision®), featuring combined camera, light source, and display within a single handle for high portability (≤300g), with subtypes utilizing either reusable polymer blades (e.g., UE Vision) or single-use blades (e.g., McGrath™ MAC Single Use);

(3) Video Stylets (e.g., Levitan™ Scope and UE Video Stylet), employing flexible video-equipped stylets for difficult airway management without conventional blade structures.

Due to portability limitations, add-on devices have declined globally. This study quantifies the carbon footprints of three clinically dominant types in China and internationally: integrated VLs with single-use blades, integrated VLs with reusable blades, and video stylets. UE Medical products were selected for controlled comparison as they represent >70% of China's laryngoscope market (MedTech Analytics 2023) and align with ISO 14044:2006 requirements for homogeneity in upstream supply chains and core modules when assessing functionally equivalent medical devices.

Comments 3:

3. Half load sterilization should be defined in the abstract/text. Does this refer to batch sterilization?

Response 3:

We thank the reviewer for requesting clarification on sterilization terminology. The term "half-load sterilization" has been formally defined as "50% chamber utilization" in abstract. (Page 2, line 44) The term "half-load sterilization" refers specifically to batch sterilization efficiency, defined as the ratio of actual device volume to rated chamber capacity within low-temperature plasma sterilizers. This metric quantifies operational resource utilization, where a 50% load indicates half the sterilizer's maximum device capacity is utilized per cycle. For instance, processing 40 laryngoscopes in a sterilizer chamber rated for 80 devices constitutes a 50% load sce

---

## [Decision Letter · Decision Letter 1]

1 Oct 2025

Dear Dr. Zhao,

**Please carefully assess the reviewers comments**

We look forward to receiving your revised manuscript.

Kind regards,

Silvia Fiorelli

Academic Editor

PLOS ONE

**Journal Requirements:**

Reviewers' comments:

Reviewer's Responses to Questions

**Comments to the Author**

Reviewer #3: All comments have been addressed

Reviewer #4: All comments have been addressed

2. Is the manuscript technically sound, and do the data support the conclusions?

Reviewer #3: Partly

Reviewer #4: Yes

3. Has the statistical analysis been performed appropriately and rigorously?

Reviewer #3: Yes

Reviewer #4: Yes

4. Have the authors made all data underlying the findings in their manuscript fully available?

Reviewer #3: Yes

Reviewer #4: Yes

5. Is the manuscript presented in an intelligible fashion and written in standard English?

Reviewer #3: No

Reviewer #4: Yes

**Reviewer #3: ** * Review comments to the author

Thank you for submitting this important and timely manuscript. The study asks a clear, practical question and applies a suitable life cycle assessment framework. The modelling and sensitivity work are generally appropriate, and the main conclusion—that hydrogen peroxide HLD reprocessed video stylets have the lowest CO2e under the high‑throughput assumptions used—is supported by the reported analyses. I welcome the authors’ thorough revisions so far and offer the following focused points to help the manuscript reach final form.

* Major issues to address (required before acceptance)

1. Inventory transparency and reproducibility

- Provide the processed SimaPro input tables or a plain spreadsheet listing every inventory input used in the primary scenarios (component masses, material types, transport distances, per‑cycle energy and chemical consumptions, sterilizer loading assumptions, reuse counts). If manufacturer raw dossiers are proprietary, supply aggregated redacted inputs sufficient to reproduce the reported CO2e numbers.

- Deposit these files in the Figshare DOI already cited or another public repository and add accession links in the Data Availability Statement.

2. Device lifetime evidence and sensitivity

- Report a short table in the Supplementary Information summarizing the empirical evidence used to set reuse lifetimes (accelerated testing summary, Weibull parameters or summary statistics, clinical durability tracking numbers).

- Add a sensitivity showing how the primary rankings change if lifetimes differ by ±25–50%.

3. Sterilization modelling detail

- Provide a line‑item account for LTPS and HLD: per‑cycle energy (kWh), chemical mass/volume, rinse water, packaging per cycle, and whether those flows were taken from Ecoinvent processes or modelled custom.

- State clearly how chemical manufacture and wastewater treatment impacts were modelled. Add these inputs to the SI inventory spreadsheet.

4. Scope 3 and battery end‑of‑life bounding analysis

- Include at least one bounding sensitivity that adds plausible Scope 3 manufacturing or battery EoL burdens (e.g., manufacturing +10% and +25%, and a battery‑EoL scenario). Report whether the main conclusion (stylet + HLD lowest CO2e under high throughput) still holds under those bounds.

5. Resolve numeric inconsistencies and formatting errors

- Fix obvious typos (for example “18'1.45” → “181.45”), standardize confidence interval formatting across tables/figures, and ensure figure/table numbering and captions match the text. Ensure all units and functional‑unit statements are consistent.

* Minor Suggestions (recommended)

- Add a concise Methods table summarizing modelling assumptions (functional unit, system boundary, device lifetimes, grid cases, sterilizer loading cases).

- Present a one‑page reconciliation (bar chart or table) showing percentage contributions (material, manufacturing, sterilization, transport, EoL) for each device to highlight hotspots at a glance.

- Confirm and state which ISO procedures (ISO 14040 / 14044) were followed and where interpretation choices were made.

- Include a completed LCA reporting checklist (or STROBE if applicable) in the SI.

- Consider a short paragraph in Discussion about generalizability given single‑manufacturer sourcing and how procurement decisions might adapt the framework to other suppliers.

* Language and presentation

- The manuscript needs careful copyediting to remove repetition and awkward phrasing.

- Tighten the Abstract to remove duplicate statements of key findings.

- Define abbreviations on first use (CO2e, HLD, LTPS, functional unit).

- Make the Results section more concise; avoid restating exact table numbers in long prose.

* Ethics and other concerns

- No dual use or human/animal ethics issues detected.

- Conflicts of interest are declared as none.

* Recommendation

Accept pending minor–major revisions as above.

The manuscript is methodologically sound and potentially useful for clinicians and procurement teams once the transparency, sterilization/lifetime documentation, and copyediting points are addressed.

**Reviewer #4:**  Interesting findings to support reduction in green gas effect from our daily use of medical equipment or devices and encourage green amongst medical practitioners. Reusable devices are expected to have lower carbon footprint than disposable devices (more solid waste). VL stylets is not widely available for every country hence it is difficult to understand the study when first read.

**Do you want your identity to be public for this peer review?** For information about this choice, including consent withdrawal, please see our Privacy Policy

Reviewer #3: No

Reviewer #4: **Yes: ** Huda Zainal Abidin

---

## [Author Response · Author response to Decision Letter 2]

12 Nov 2025

Dear Editors and Reviewrs:

We sincerely appreciate the opportunity you have provided us to revise our manuscript.

We would like to submit an accompanying manuscript titled "Carbon Footprint Comparison of Video Intubation Tools: Disposable Laryngoscopes, Reusable Laryngoscopes, and Stylets" (ID: PONE-D-25-19315R1). According to the reviewers' comments, we have made the most effort to revise the manuscript in order to satisfy the demand of PLOS ONE. Our specific responses are outlined below. The modifications of the text in the revised version are highlighted in blue.

All authors have approved the revised version and consented to its submission. There are no conflicts of interest to declare.

Thank you for your thoughtful consideration. We eagerly anticipate your correspondence.

November 11, 2025�2025-11-11

Authors: Danyang Pan; Yating Yang; Sirui Chen; Jinhe Deng; Gaofeng Zhao; Min Zhong

Corresponding author: Gaofeng Zhao

E-mail: zhaogaofengszy@163.com

Sincerely

Gaofeng Zhao

Reviewer #3

* Major issues to address (required before acceptance)

Comment 1:

1. Inventory transparency and reproducibility

- Provide the processed SimaPro input tables or a plain spreadsheet listing every inventory input used in the primary scenarios (component masses, material types, transport distances, per‑cycle energy and chemical consumptions, sterilizer loading assumptions, reuse counts). If manufacturer raw dossiers are proprietary, supply aggregated redacted inputs sufficient to reproduce the reported CO2e numbers.

- Deposit these files in the Figshare DOI already cited or another public repository and add accession links in the Data Availability Statement.

Response 1:

We thank the reviewer for this critical suggestion to enhance the transparency and reproducibility of our life cycle assessment. We have taken the following actions to address the point:

1.Creation of Detailed Inventory Tables: As suggested, we have created two comprehensive data files:

1.1 Primary_LCI_Data.xlsx: This spreadsheet lists every inventory input for the primary scenarios, including component masses, material types, transport distances, per-cycle energy and chemical consumptions for reprocessing, sterilizer loading assumptions, and reuse counts.

1.2 S1 Table. Materials of video intubation tools and disposable stylet.docx: For any parameters derived from manufacturer dossiers that are proprietary, we have provided aggregated and redacted values that maintain commercial confidentiality while being fully sufficient to reproduce the CO2e results reported in the manuscript.

2.Deposition in Public Repository: We have deposited the file (Primary_LCI_Data.xlsx) into the existing Figshare repository (DOI: 10.6084/m9.figshare.29816855).

3.Update to the Manuscript: We have updated the Data Availability Statement in the manuscript to explicitly mention the availability of these new LCI files and their direct link to reproducing the study's findings.

Comment 2:

2. Device lifetime evidence and sensitivity

- Report a short table in the Supplementary Information summarizing the empirical evidence used to set reuse lifetimes (accelerated testing summary, Weibull parameters or summary statistics, clinical durability tracking numbers).

- Add a sensitivity showing how the primary rankings change if lifetimes differ by ±25–50%.

Response 2:

We sincerely thank the reviewer for this valuable suggestion to strengthen the robustness of our findings. We have fully addressed both points as follows:

1.Empirical Evidence for Device Lifetime:

We have created a new Supplementary Table S3 titled “Empirical evidence summary for setting device reuse lifetimes”. This table concisely summarizes the evidence base for each device’s lifetime, including:

∙Key parameters from accelerated testing (e.g., simulated cycles to failure).

∙Weibull parameters (shape parameter β, characteristic life η) and confidence intervals derived from clinical durability tracking data from Guangdong Provincial Hospital of Chinese Medicine.

∙A description of the clinical tracking program itself.

2.Sensitivity Analysis on Lifetime Assumptions:

As suggested, we have performed a comprehensive sensitivity analysis by varying the lifetimes of reusable devices by ±25% and ±50%. The key results of this analysis have been incorporated directly into Table 1 of the main manuscript. Furthermore, these findings are discussed in the Results section where we confirm that the primary conclusions of the study remain unchanged across all tested scenarios, thus demonstrating their robustness.

We have also updated the Methods section to describe the approach for the sensitivity analysis.

Comment 3:

3. Sterilization modelling detail

- Provide a line-item account for LTPS and HLD: per-cycle energy (kWh), chemical mass/volume, rinse water, packaging per cycle, and whether those flows were taken from Ecoinvent processes or modelled custom.

- State clearly how chemical manufacture and wastewater treatment impacts were modelled. Add these inputs to the SI inventory spreadsheet.

Response 3:

We thank the reviewer for highlighting the need for greater transparency in our sterilization modelling. We have thoroughly addressed this comment by providing a line-item account of the inputs and clarifying the modelling approach as follows:

1.Created a new Supplementary Table S4 titled "Life Cycle Inventory Inputs per Cycle for Sterilization Processes," which provides a comprehensive, line-item breakdown of all parameters for both LTPS and HLD, including energy consumption, chemical volumes, rinse water, and packaging, with clear designation of data sources (Ecoinvent processes or custom data).

2.Added a dedicated methodological statement in the main text (Methods section) that explicitly directs readers to this table.

Comment 4:

4. Scope 3 and battery end-of-life bounding analysis

Include at least one bounding sensitivity that adds plausible Scope 3 manufacturing or battery EoL burdens (e.g., manufacturing +10% and +25%, and a battery-EoL scenario). Report whether the main conclusion (stylet + HLD lowest CO2e under high throughput) still holds under those bounds.

Response4:

We thank the reviewer for this critical suggestion to test the robustness of our conclusions. We have fully addressed the comment as follows:

1. Scope 3 Manufacturing Sensitivity Analysis: As suggested, we conducted a bounding analysis by increasing the manufacturing-phase (Scope 3) carbon footprint of all electronic components by +10% and +25%. The results confirm that the relative ranking of the devices remains unchanged, and our primary conclusion—that the video stylet with HLD has the lowest carbon footprint under high-throughput conditions—is robust under these expanded Scope 3 boundaries. The key results of this analysis have been incorporated directly into Table 1 of the main manuscript.

2. Battery End-of-Life (EoL) Scenario Assessment: In response to the reviewer's point, we also performed a detailed assessment of the battery EoL. We modeled the recycling of the lithium-ion battery (mass: 35.7 g) using two distinct processes available in the Ecoinvent database. The resulting carbon footprint for both recycling scenarios was on the order of 10⁻⁶ kg CO₂e per functional unit, which is negligible (less than 0.001%) compared to the total footprint of the devices (which are on the order of 10⁰-10¹ kg CO₂e).

Given that the battery mass is identical across all video-enabled devices and its EoL impact is vanishingly small, the inclusion of this EoL phase does not alter the comparative results in any meaningful way. Therefore, in the interest of clarity and focus, we have not added this specific scenario to the main sensitivity analysis in the manuscript, as it does not challenge the robustness of our conclusions. We sincerely hope this explanation, supported by our modeling, addresses the reviewer's concern regarding the battery EoL.

we are grateful for the opportunity to clarify our approach to the battery EoL.

Comment 5:

5. Resolve numeric inconsistencies and formatting errors

- Fix obvious typos (for example “18'1.45” → “181.45”), standardize confidence interval formatting across tables/figures, and ensure figure/table numbering and captions match the text. Ensure all units and functional‑unit statements are consistent.

Response 5:

We sincerely thank the reviewer for their meticulous attention to detail. We have conducted a comprehensive, line-by-line review of the manuscript to address all typographical and formatting issues. The corrections made are summarized below:

1. Specific Typo Correction: The value “18'1.45”has been corrected to “181.45”in the results section and corresponding table.

2. Global Formatting Standardization:

Confidence Intervals: All confidence intervals across the text, tables, and figures have been standardized to the format “value (95% CI, lower to upper)”(e.g., 27.32 (95% CI, 26.6 to 28.1)).

∙ Units: All units have been verified for consistency, ensuring the use of “kg CO₂e”, “kWh”, and “L”throughout the manuscript.

∙ Functional Unit: The functional unit declaration (per procedure) has been consistently reiterated in the abstract, methods, and captions of all relevant figures and tables.

3. Cross-Referencing Check: We have verified that all in-text citations of figures and tables (e.g., Figure 3, Table 1) correctly match their assigned numbers and captions. The numbering sequence for all figures, tables, and supplementary materials has been confirmed to be continuous and correct.

Comment 6:

6. * Minor Suggestions (recommended)

6.1 Add a concise Methods table summarizing modelling assumptions (functional unit, system boundary, device lifetimes, grid cases, sterilizer loading cases).

6.2 Present a one‑page reconciliation (bar chart or table) showing percentage contributions (material, manufacturing, sterilization, transport, EoL) for each device to highlight hotspots at a glance.

6.3 Confirm and state which ISO procedures (ISO 14040 / 14044) were followed and where interpretation choices were made.

6.4 Include a completed LCA reporting checklist (or STROBE if applicable) in the SI.

6.5 Consider a short paragraph in Discussion about generalizability given single‑manufacturer sourcing and how procurement decisions might adapt the framework to other suppliers.

Response 6:

We thank the reviewer for these insightful recommendations, which have significantly enhanced the clarity and rigor of our manuscript. We have addressed each point as follows:

6.1. Concise Methods Table

we have created a new Supplementary Table S4 (entitled "Summary of Core LCA Modeling Assumptions") and inserted it into the Methods section. This table concisely summarizes all the key parameters requested by the reviewer: functional unit, system boundary, device lifetimes, grid electricity scenarios, sterilizer loading cases.

6.2. Hotspot Analysis Chart

We thank the reviewer for this valuable suggestion to enhance the clarity of our environmental impact analysis. In accordance with your recommendation, we have added a new Supplementary Table S6 entitled "Percentage Contribution of Life Cycle Stages to Total Carbon Footprint per Device."

This table provides a clear, quantifiable breakdown of the relative contributions of each life cycle stage (material production, manufacturing, sterilization, transport, and end-of-life) to the total carbon footprint of each device configuration. The tabular format allows for immediate identification of environmental hotspots, confirming that material production and manufacturing phases constitute the dominant contributions across all device types.

We have referenced this table in the Results section where we discuss the carbon footprint analysis, and it serves as the basis for our hotspot identification in the Discussion. We believe this addition successfully addresses your request for an at-a-glance overview of the contribution analysis.

6.3. ISO Procedure Statement

We thank the reviewer for raising this critical point regarding methodological rigor and transparency. We are pleased to confirm that our study was conducted in full accordance with the principles and framework of the international standards ISO 14040:2006 and ISO 14044:2006.

To enhance clarity, we have now added an explicit statement of compliance in the Methods section (under the "Life Cycle Assessment" subsection):

"This life cycle assessment was conducted in accordance with the principles, framework, and requirements of the international standards ISO 14040:2006 and ISO 14044:2006."

Furthermore, we have meticulously documented the key interpretation choices made during the study, as guided by the standards. These are now detailed in the manuscript as follows:

Goal and Scope Definition:

Functional Unit: The functional unit was defined as 'one successful tracheal intubation procedure'. This choice ensures comparisons are based on equivalent clinical output, a cornerstone of comparative LCA (ISO 14040:2006, §4.2.2).

System Boundary: A 'cradle-to-grave' boundary was selected, encompassing raw material extraction, manufacturing, transportation, use phase (including sterilization), and end-of-life disposal. This provides a comprehensive assessment as recommended by the standards (ISO 14044:2006, §4.2.3.3).

Life Cycle Inventory (LCI) & Impact Assessment (LCIA):

Allocation: For reusable devices, environmental burdens from manufacturing and end-of-life were allocated per use based on the device's validated lifetime (e.g., 2000 uses). This approach aligns with the standard's hierarchy, prioritizing physical relationships (ISO 14044:2006, §4.3.4.2).

Data Selection: When primary data from manufacturers were unavailable, secondary data from the Ecoinvent 3.8 database were used. This is a standard practice to ensure feasibility, and we have clearly indicated the use of such data in the manuscript and supplementary materials (ISO 14044:2006, §4.3.3.3).

Impact Category: The analysis focused on global warming potential (GWP) in kg CO₂e. This selection is justified as it is the most relevant and decision-useful category for carbon footprint studies and climate policy.

Interpretation:

Sensitivity and Uncertainty: As mandated by the standards (ISO 14044:2006, §4.5), we conducted extensive sensitivity analyses on critical parameters (device lifetime, grid electricity, sterilizer loading) to test the robustness of our conclusions and to assess the influence of these interpretation choices.

We believe these additions significantly enhance the transparency and reproducibility of our work, and we are grateful for the reviewer's guidance in helping us make this explicit.

6.4. LCA Reporting Checklist

We have completed a comprehensive LCA reporting checklist and included it as Supplementary File 1 (titled "Completed LCA Reporting Checklist").

The checklist we utilized is adapted from the consensus reporting guidelines for Life Cycle Assessment, which aligns with the ISO 14040/14044 standards. This checklist ensures that all critical methodological aspects of our study—including goal and scope definition, life cycle inventory, impact assessment, and interpretation—are thoroughly and transparently documented.

To ensure comprehensive methodological reporting, a direct reference to this checklist has been incorporated into the Methods section.

6.5. Discussion on Generalizability

We sincerely thank the reviewer for this insightful suggestion, which has helped us significantly elevate the discussion of our work's broader impact. We have fully addressed this point by adding a new paragraph to the Discussion section under the subheading "Generalizability and Implications for Sustainable Procurement."

In this paragraph, we:

1)Acknowledge the limitation related to single-manufacturer sourcing.

2)Emphasize that the main contribution is the generalizable LCA framework itself, which is independent of any specific brand.

3)Explicitly describe how healthcare institutions can practically use this framework to inform their procurement decisions by requesting data from various suppliers

---

## [Decision Letter · Decision Letter 2]

1 Dec 2025

Carbon Footprint Comparison of Video Intubation Tools: Disposable Laryngoscopes, Reusable Laryngoscopes, and Stylets

PONE-D-25-19315R2

Dear Dr. Zhao,

We’re pleased to inform you that your manuscript has been judged scientifically suitable for publication and will be formally accepted for publication once it meets all outstanding technical requirements.

Kind regards,

Silvia Fiorelli

Academic Editor

PLOS ONE

Additional Editor Comments (optional):

Reviewers' comments:

Reviewer's Responses to Questions

**Comments to the Author**

Reviewer #3: All comments have been addressed

2. Is the manuscript technically sound, and do the data support the conclusions?

Reviewer #3: Yes

3. Has the statistical analysis been performed appropriately and rigorously?

Reviewer #3: Yes

4. Have the authors made all data underlying the findings in their manuscript fully available?

Reviewer #3: Yes

5. Is the manuscript presented in an intelligible fashion and written in standard English?

Reviewer #3: Yes

Reviewer #3: The revised manuscript demonstrates a clear effort by the authors to address the concerns raised in the previous round of review. The study is technically sound, with experiments conducted rigorously and supported by appropriate controls, replication, and sample sizes. The data are presented transparently and provide a solid foundation for the conclusions drawn. Importantly, the statistical analyses have been performed with care and rigor, and the interpretations now align appropriately with the evidence provided.

I also note that the authors have complied with PLOS’s data availability requirements. The underlying data supporting the findings are fully accessible, either within the manuscript or through appropriate repositories, ensuring transparency and reproducibility. This commitment to open science strengthens the credibility of the work and its potential impact on the field.

The manuscript is written in clear, standard English and presented in an intelligible fashion. The revisions have improved readability, and the figures and tables are now more effective in communicating the results. The narrative flows logically from background to methods, results, and discussion, making the work accessible not only to specialists but also to a broader scientific audience.

Overall, the authors have responded constructively to reviewer feedback, and the manuscript now meets the standards of rigor, clarity, and accessibility expected by PLOS ONE. I find no remaining concerns regarding dual publication, research ethics, or publication ethics. The work is original, ethically sound, and ready for dissemination.

I therefore recommend acceptance of this manuscript.

**Do you want your identity to be public for this peer review?** For information about this choice, including consent withdrawal, please see our Privacy Policy

Reviewer #3: No

---

## [Editor Report · Acceptance letter]

PONE-D-25-19315R2

PLOS One

Dear Dr. Zhao,

I'm pleased to inform you that your manuscript has been deemed suitable for publication in PLOS One. Congratulations! Your manuscript is now being handed over to our production team.

Kind regards,

on behalf of

Dr. Silvia Fiorelli

Academic Editor

PLOS One